# Inhibitory Effect of Berberine on Broiler P-glycoprotein Expression and Function: In Situ and In Vitro Studies

**DOI:** 10.3390/ijms20081966

**Published:** 2019-04-22

**Authors:** Yujuan Zhang, Li Guo, Jinhu Huang, Yong Sun, Fang He, Mire Zloh, Liping Wang

**Affiliations:** 1MOE Joint International Research Laboratory of Animal Health and Food Safety, College of Veterinary Medicine, Nanjing Agricultural University, Nanjing 210095, China; 2016207009@njau.edu.cn (Y.Z.); 2014107023@njau.edu.cn (L.G.); jhuang@njau.edu.cn (J.H.); ljm102100@163.com (Y.S.); 2015107025@njau.edu.cn (F.H.); 2Faculty of Pharmacy, University Business Academy, Trg mladenaca 5, 21000 Novi Sad, Serbia

**Keywords:** berberine, P-glycoprotein, inhibition, MDCK-chAbcb1 cell line, chicken

## Abstract

Overcoming P-glycoprotein (P-gp) efflux is a strategy to improve the absorption and pharmacokinetics of its substrate drugs. Berberine inhibits P-gp and thereby increases the bioavailability of the P-gp substrate digoxin in rodents. However, the effects of berberine on P-gp in chickens are still unclear. Here, we studied the role of berberine in modulating broilers P-gp expression and function through both in situ and in vitro models. In addition, molecular docking was applied to analyze the interactions of berberine with P-gp as well as with chicken xenobiotic receptor (CXR). The results showed that the mRNA expression levels of chicken P-gp and CXR decreased in the ileum following exposure to berberine. The absorption rate constant of rhodamine 123 increased after berberine treatment, as detected using an in situ single-pass intestinal perfusion model. Efflux ratios of P-gp substrates (tilmicosin, ciprofloxacin, clindamycin, ampicillin, and enrofloxacin) decreased and the apparent permeability coefficients increased after co-incubation with berberine in MDCK-chAbcb1 cell models. Bidirectional assay results showed that berberine could be transported by chicken P-gp with a transport ratio of 4.20, and this was attenuated by verapamil (an inhibitor of P-gp), which resulted in a ratio of 1.13. Molecular docking revealed that berberine could form favorable interactions with the binding pockets of both CXR and P-gp, with docking scores of −7.8 and −9.5 kcal/mol, respectively. These results indicate that berberine is a substrate of chicken P-gp and down-regulates P-gp expression in chicken tissues, thereby increasing the absorption of P-gp substrates. Our findings suggest that berberine increases the bioavailability of other drugs and that drug-drug interactions should be considered when it is co-administered with other P-gp substrates with narrow therapeutic windows.

## 1. Introduction

Berberine is a yellow isoquinoline alkaloid isolated from many plant families [1]. It has been used since the 1950s as an effective medication for the treatment of gastro-enteritis and secretory diarrhea, with safety in humans [2]. It is also effective for the prevention and treatment of diarrhea in animals, such as chickens [3]. However, studies have shown that berberine modulates the activity of P-glycoprotein (P-gp) and could mediate drug–drug interactions when it is co-administered with P-gp substrates [4,5].

P-gp, encoded by the *Abcb1* gene, belongs to the family of adenosine triphosphate (ATP)-binding cassette transporters and is usually localized in excretory and barrier-function tissues, such as the kidney and intestine [6,7]. P-gp can utilize the hydrolysis of ATP to energize the efflux of a broad range of substrates, including commonly used antimicrobial agents licensed in veterinary medicine (e.g., ivermectin, enrofloxacin, and danofloxacin) [8,9,10], influencing the absorption, distribution, and excretion of substrates [11]. Therefore, overcoming P-gp efflux is a strategy to improve the absorption and pharmacokinetics of substrates.

Berberine dose-dependently increases the bioavailability of digoxin and cyclosporine A, two well-known P-gp substrates, by inhibiting intestinal P-gp [4]. In contrast, berberine induces the expression of P-gp in human intestinal and liver cells [12]. Furthermore, the effect of berberine on P-gp expression depends on the cell line [13]. These findings suggest that berberine can regulate P-gp expression and can consequently affect the pharmacokinetics of therapeutic agents. However, little is known about the role of berberine in modulating chicken P-gp expression and activity.

This study aimed to determine the effects of berberine on chicken P-gp expression and functional activity. Our results indicate that berberine is a substrate of chicken P-gp and could down-regulate the expression and efflux activity of chicken P-gp, thus increasing the absorption of P-gp substrates, as verified by in situ and in vitro experiments. These findings are informative for the rational use of drugs in the poultry industry to increase bioavailability or avoid adverse effects; accordingly, this study has practical applications for therapeutic efficacy and food safety.

## 2. Results

### 2.1. Effect of Berberine on Abcb1 and CXR mRNA Expression in Broilers

The mRNA expression levels of *Abcb1* and chicken xenobiotic receptor (*CXR*) were evaluated in the ileum of broilers by real-time RT-PCR after berberine administration at various concentrations (40 and 80 mg/kg) for 24 h (Figure 1). Compared with levels in the controls, the two concentrations of berberine significantly down-regulated *Abcb1* and *CXR* in the ileum at all time-points (*p* < 0.01). In addition, the decreased mRNA levels of *Abcb1* were in line with the decreased *CXR* mRNA levels.

### 2.2. Berberine Affected Rho123 Uptake in the Jejunum of Broilers by In Situ Perfusion

A perfusion model in the jejunum of broilers was chosen for the drug–drug interaction study (berberine–substrate) to further evaluate whether down-regulated *Abcb1* mRNA expression by berberine is accompanied by a weaker transport function in the small intestine. The jejunum perfusion was evaluated by monitoring the Rho123 concentration (control and berberine-treated samples) over time, as depicted in Figure 2. *K*_a_ and *P*_app_ values for Rho123 in the jejunum of broilers are presented in Table 1. The results showed that treatment with the two concentrations of berberine (40 and 80 mg/kg) for 24 h both resulted in an obvious decrease (*p* < 0.05) in Rho123 concentrations in the perfusion fluid (Figure 2) and a significant increase in the Rho123 absorption rate constant (*K*_a_) (*p* < 0.05) (Table 1).

### 2.3. Effect of Berberine on P-gp Medicated Rho123 Efflux in MDCK-chAbcb1 Cells

We investigated the effect of berberine on P-gp function in MDCK-chAbcb1 cells with Rho123 as a probe substrate. For this, a monolayer of MDCK-chAbcb1 cells was pretreated with berberine at various concentrations (5, 20, and 40 µM) for 2 or 8 h. There was no significant difference in Apical to Basolateral (AP→BL) transported Rho123 among berberine concentrations (Figure 3A,C). However, the addition of berberine to the cell incubation mixture resulted in less Rho123 transport to the apical side from the basolateral side (Figure 3B,D), suggesting that the polarized efflux of Rho123 was inhibited by berberine in MDCK-chAbcb1 cells. To quantify the inhibitory effect of berberine, the efflux ratio (ER) values and inhibition rates were calculated, as shown in Table 2. We found that 2 h treatment with berberine reduced the ER of Rho123 in a concentration-dependent manner. In particular, treatment with 5, 20, and 40 μM berberine for 2 h significantly decreased the ERs of Rho123 from 7.28 to 4.26, 3.4 (*p* < 0.05), and 1.92 (*p* < 0.01), with inhibition rates of 41.48%, 53.3%, and 73.6%, respectively. Extending the berberine treatment time to 8 h did not result in further decreases in the ERs and inhibition rates, indicating that the inhibitory effect of berberine on chicken P-gp depends on the concentration but not the treatment time. Therefore, 2 h of incubation was selected for further analyses of verapamil. Similarly, in verapamil-treated cells, there was no significant difference in AP→BL transported Rho123 (Figure 4A). However, less Rho123 was transported to the apical side from the basolateral side (Figure 4B). The ER decreased gradually as the verapamil concentration increased (Table 3). High concentrations of verapamil (100, 1000, and 5 000 μM) were related to an excellent ability to inhibit P-gp function, significantly reducing the ER to 3.51 (*p* < 0.05), 1.94 (*p* < 0.05), and 1.37 (*p* < 0.01), compared with the ER (6.24) of the control. The rates of inhibition increased to 43.75%, 68.91%, and 78.04%, respectively.

### 2.4. Transporting Interactions of Berberine with Veterinary Drugs in MDCK-chAbcb1 Cells

The inhibitory effect of berberine on CP-gp function was evaluated using additional veterinary medicines, which were identified as substrates of CP-gp in our previous study [14]. For this, the effects of berberine (5, 20, and 40 µM) on the directional transport of tilmicosin, ciprofloxacin, clindamycin, sulfadiazine, ampicillin, and enrofloxacin were examined. For tilmicosin and ciprofloxacin transport, berberine treatment significantly reduced the ER (*p* < 0.01) in a concentration-dependent manner compared with the ER of the control (Figure 5A,B). In the absence of berberine, the ER for tilmicosin was 4.26. In contrast, in the presence of 5, 20, and 40 μM berberine, the ER values were reduced to 1.20, 0.91, and 0.79, respectively, suggesting a strong inhibitory effect on P-gp transport by berberine. Similarly, berberine also reduced ciprofloxacin transport; the ER for the control treatment was 2.43, while the ER values for treatment with 5, 20, and 40 μM berberine were 0.77, 0.75, and 0.70, respectively.

The transport of clindamycin was significantly inhibited (*p* < 0.05) by berberine; the ER values for 5 and 20 μM berberine were 4.70 and 3.79, respectively, compared with 5.67 in the absence of berberine (Figure 5C). When the concentration of berberine increased from 20 μM to 40 μM, the transport ratio became 3.84.

For sulfadiazine, berberine produced a biphasic response to P-gp-mediated transport, i.e., stimulation at low concentrations and inhibition at high concentrations (Figure 5D). Under berberine-free conditions, the ER was 2.77. However, in the presence of 5 μM berberine, the ER increased significantly to 3.71 (*p* < 0.05). Increasing the berberine concentration to 20 μM did not obviously alter the ER. However, when the berberine concentration was increased to 40 μM, the ER decreased significantly to 1.93 (*p* < 0.05).

The transport of ampicillin or enrofloxacin in MDCK-chAbcb1 cells was not substantially influenced by berberine (Figure 5E,F). Compared with the control, 40 μM berberine significantly decreased the ER from 2.54 to 2.17 for ampicillin and from 2.01 to 1.35 for enrofloxacin (*p* < 0.05).

We also found that berberine significantly inhibited the BL→AP transport of substrates and enhanced the apical-to-basolateral flux (AP→BL) to an extent. Furthermore, there were no significant changes in the ER and *P*_app_ for all of the test drugs in MDCK cells (Appendix A). These results suggest that berberine could inhibit the transport function of CP-gp, thus increasing the permeability of its substrates.

### 2.5. Berberine Is a Substrate for Chicken P-gp Proved Based on a Bidirectional Transport Assay in MDCK chAbcb1 Cells

To further determine how berberine affects the function of P-gp, we hypothesized that berberine is a substrate of P-gp. To evaluate this hypothesis, a bidirectional transport assay of berberine was performed using MDCK and MDCK-chAbcb1 cells. In MDCK parental cells, the berberine transport ratio was 1.24 (Figure 6A). However, in MDCK-chAbcb1 cells, the berberine transport ratio was 4.2 due to increased apically and decreased basolaterally directed berberine translocation as compared to those of the parental cell line (Figure 6B). This transport was abolished by the specific P-gp inhibitor verapamil (100 μM), resulting in a reduced transport ratio of 1.13 in MDCK-chAbcb1 cells that was similar to the transport ratio of 1.21 in MDCK parental cells (Figure 6C,D). The above data indicate that chicken P-gp was actively involved in berberine transport.

### 2.6. Berberine might Favorably Interact with P-gp and CXR, as Analyzed by Molecular Docking Modeling

The initial three-dimensional structure of CP-gp was obtained using the homology modeling software ITASSER (Figure 7). The model was further refined by a molecular dynamics simulation of the fully solvated initial structure embedded in the lipid membrane and ligand binding pocket. The final frame of the 1-ns simulation trajectory was selected for validation and further docking studies. The protein structure without the solvent and membrane was compared to the experimental structure of mouse P-gp (PDB ID: 3G60) by a structural protein alignment of their binding sites. The sequence alignment of the binding sites is shown in Appendix A. The sequence identity was 80% in the putative binding site of the homology model (Appendix A). The binding site was further evaluated by docking a set of selected P-gp substrates and inhibitors, showing consistent and specific patterns for substrate binding when compared to inhibitors (Table 4, Appendix A).

The previously reported homology model of CXR [15] was used to dock a set of small molecules and evaluate their affinity to the binding site of the ligand binding domain. The docking scores and poses agreed with the previous results, and the binding energy of −7.8 kcal/mol suggested that berberine interacts favorably with CXR (Table 4, Figure 8A). Out of 13 residues that interact with berberine in the docking pose (Figure 8B,C), 4 aromatic residues, F204, F249, W260, and F277 (Figure 8D), could be key sites responsible for the favorable binding, as they were observed in more than 50% of frames in molecular dynamics simulations.

The docking pose of berberine (Figure 9A) and the score for the binding of berberine to CP-gp (−9.5 kcal/mol) indicated a higher affinity of berberine to the efflux pump than to CXR (Table 4). The position of berberine and residues that interact with CP-gp binding sites (Y315, F344, and F986, as depicted in Figure 9B,C) were consistent with the residues predicted by docking to be involved in berberine binding to mouse or human P-gp structures [13,16,17]. However, the molecular dynamics simulation of the CP-gp/berberine complex indicated that the key residues responsible for the dynamic interaction with berberine were F311, Y315, F991, and, importantly, Y318 (Figure 9D).

For the comparative analysis, the docking of all small molecules was evaluated against two other efflux pumps in chickens, CABCC7 and CMATE transporters (Appendix A). It is apparent that favorable interactions can form between all studied small molecules and these two transporters. Specifically, the docking scores for berberine against both transporters were −7.9 kcal/mol in both cases. These values indicate less favorable interactions when compared with the docking score of berberine against CP-gp (−9.5 kcal/mol). These results suggest that although there is a possibility that berberine could be transported by other transporters, it is more likely that berberine transport is mediated by P-gp in chickens. The docking scores for efflux pump inhibitors against P-gp were lower than those obtained in docking studies of the same inhibitors against the CABCC7 and CMATE transporters. Therefore, the inhibitory effects may be associated with the suppression of berberine transport by CP-gp, further supporting the experimental observation that berberine efflux is likely specific to CP-gp.

## 3. Discussion

Berberine effectively controls coccidial infection and Clostridium perfringens-associated necrotic enteritis in broilers [18,19,20]. In addition, berberine is a good alternative feed additive and has been used safely in starter, grower, and finisher feeds for broilers, as the commercial poultry industry has faced increasing pressure to reduce the use of antimicrobial growth promoters. However, berberine may affect the bioavailability or the withdrawal period of P-gp substrates in rats [4]. The results of this study suggest that berberine is a substrate of chicken P-gp and could down-regulate the expression and activity of P-gp in broilers, thus increasing the absorption of P-gp substrates both in situ and in vitro. To our knowledge, this is the first report of the regulatory effect of berberine on P-gp in chickens and the underlying mechanism using both in situ and in vitro models.

In this study, we found that berberine down-regulated the expression of *Abcb1* in chickens. Though the protein level of P-gp was not analyzed owing to a lack of antibodies to chicken P-gp, the decreased efflux activity of chicken intestinal P-gp was further confirmed by an in situ experiment using Rho123 as a probe substrate. It has been reported that various flavonoids, including berberine, regulate P-gp expression via pregnane X receptor (PXR) and constitutive androstane receptor (CAR) in humans [21]. CXR, a chicken xenobiotic-sensing orphan nuclear receptor, is related to both PXR and CAR [22]. Our research showed that berberine might regulate chicken *Abcb1* expression via CXR, as the *Abcb1* and *CXR* genes displayed similar changes in response to berberine. Our results are very similar to the finding that camptothecin (an isoquinoline alkaloid, like berberine) attenuates cytochrome P450 3A4 induction by blocking the activation of human PXR [23]. However, The results of this study are inconsistent with the recent data showing that berberine up-regulate P-gp through transcription factor nuclear factor erythroid 2-related factor 2 (Nrf2) but not through PXR when colitis occurs in rats [24]. This discrepancy is most likely due to the different regulatory pathways of P-gp between normal and pathologic systems. Therefore, berberine regulate the *Abcb1* gene expression through Nrf2-mediated pathway when colitis occurs, however, may through PXR-mediated pathway in a healthy organism. On the other hand, other studies showed that there are species-specific variations in *Abcb1* gene regulation. These studies provided evidence implicating complex mechanisms for regulation of the *Abcb1* gene [25,26,27] Furthermore, molecular docking showed that berberine could form favorable interactions with the binding pocket of CXR, with a docking score of −7.8 kcal/mol. In addition, CXR could bind to chicken RXRɤ, forming a heterodimer, and then bind to an identified phenobarbital-responsive unit to regulate the expression of chicken CYP2H1, suggesting an essential role of CXR in gene regulation [28]. We hypothesize that berberine cannot enter the nucleus alone; accordingly, berberine itself should not have any effect on the DNA/gene sequences that impact P-gp promoter transcription. However, the effect of berberine on P-gp expression may occur through the nuclear receptor, which is shuttled back and forth between the caryoplasm and the cytoplasm [29,30]. Of course, the precise relationship between *CXR* and *Abcb1* should be studied further and confirmed in vitro.

The inhibitory effect of 40 μM berberine on chicken P-gp was comparable with that of 5000 μM verapamil (inhibition rates, 73.63% vs. 78.04%). This indicates that berberine is a potent P-gp inhibitor. Our results also suggest that berberine is an effective P-gp substrate with a high transport ratio in MDCK-chAbcb1 and a low docking score, in good agreement with previous results in humans [31]. In addition, we observed that berberine treatment reduces the ER of Rho123 in a concentration-dependent manner. Therefore, we propose that berberine may competitively inhibit chicken P-gp when co-administered with other P-gp substrates.

Tilmicosin, ciprofloxacin, clindamycin, sulfadiazine, ampicillin, and enrofloxacin are all antimicrobial agents with bacteriostatic activity and are commonly used in the poultry industry. We previously demonstrated that all six drugs are substrates of chicken P-gp [14]. The antimicrobial effects of most antibiotics are partly dependent on their blood concentrations. Our results indicated that berberine enhanced the apparent permeability coefficients of these antimicrobial agents by inhibiting the efflux function of P-gp in MDCK-chAbcb1 cells. Our previous in vivo experiments have also shown that berberine increases the bioavailability of orally administered enrofloxacin by promoting its intestinal absorption and restricting liver/kidney clearance in broilers (data published in a Chinese journal, as shown in Appendix A). However, a stimulatory effect of 5 μM berberine on chicken P-gp was observed for sulfadiazine transport. This could be explained by the biphasic effect of berberine (stimulation at low concentrations and inhibition at high concentrations) on P-gp-dependent ATPase [32]. Similarly, there is a biphasic effect of quercetin (another flavonoid) on the blood–brain barrier transport of vincristine [33]. Therefore, combining these previous findings with our experimental results, we speculate that the ultimate effects of berberine on P-gp are comprehensive, including the inhibition of P-gp gene expression, competition with other P-gp substrates, and an effect on P-gp-dependent ATPase. Nevertheless, our data indicate that berberine could inhibit the function of chicken P-gp at high concentrations (40 μM in this study) for all of the tested drugs. It has been reported that the P-gp substrates also affect P-gp ATPase activity, thus influencing the transport function of P-gp [34], which may explain the different inhibitory effects of berberine on P-gp in the transport of different drugs. Overall, a drug (e.g., tilmicosin) with a low apparent permeability was more greatly affected by berberine than a drug (e.g., enrofloxacin) with high membrane permeability. These results also indicate that a P-gp substrate drug with a low apparent permeability could be combined with berberine and to improve its bioavailability in chickens.

## 4. Materials and Methods

### 4.1. Reagents

Hygromycin B, rhodamine 123 (Rho123), verapamil, and berberine were obtained from Aladdin (Cambridge, MA, USA). Tilmicosin, ciprofloxacin, clindamycin, sulfadiazine, ampicillin, and enrofloxacin were kind gifts from the China Institute of Veterinary Drug Control (Beijing, China). All other chemicals were of analytical grade and obtained from local suppliers, unless otherwise mentioned.

### 4.2. Cell Line and Animals

The MDCK-chAbcb1 cell line stably expressing chicken P-gp was previously established in our laboratory. Briefly, the full-length chicken cDNA was first cloned into the pcDNA3.1 vector and then transfected into MDCK cells using Lipofectamine 2000. Single cells stably expressing chicken P-gp were collected using the limited dilution method in medium containing hygromycin B (100 μg/mL) to obtain the MDCK-chAbcb1 cell line [35]. The cell line was cultured in Dulbecco’s modified Eagle’s medium containing 10% fetal bovine serum, antibiotics (100 IU/mL penicillin and 80 IU/mL streptomycin), and 50 μg/mL hygromycin B and incubated at 37 °C in humidified air with 5% CO_2_.

Arbor Acres broilers (one day old) were purchased from the Liuhe Hatchery (Nanjing, Jiangsu, China). The broilers were provided a basal diet and water ad libitum and managed under the recommended humidity and temperature for 6 weeks. Then, 18 healthy broilers with similar body weight (b.w.) were randomly divided into three groups (6 broilers/group). Group I was administered phosphate-buffered saline for 24 h (control). Groups II and III were given 40 and 80 mg/kg b.w. berberine orally, respectively, and treated for 24 h. Chicken treatment procedures were approved by the Science and Technology Agency of Jiangsu Province (approval no. 2017-0007, approval date: 15th, February, 2017) and performed in accordance with the guidelines of the Science and Technology Agency of Jiangsu Province and Nanjing Agricultural University.

### 4.3. Effect of Berberine on Abcb1 and Chicken Xenobiotic Receptor mRNA Expression in the Ileum of Broilers Determined by Real-Time RT-PCR

Broilers were divided into three experimental groups and drugs were administered as described above. At the end of experiments, animals were killed by decapitation. The ileum samples were collected immediately and stored at −80 °C for further tests. Total RNA was isolated using TRIzol Reagent (Invitrogen, Carlsbad, CA, USA) according to the manufacturer’s instructions and treated with DNase to remove DNA contamination. RNAs were quantified using a photometer (Eppendorf, Hamburg, Germany) at 260/280 nm. RNA integrity was confirmed by 1% agarose gel electrophoresis and ethidium bromide staining. Then, cDNAs were synthesized using a kit (Promega, Madison, WI, USA) according to the manufacturer’s protocol. Chicken *Abcb1* and chicken xenobiotic receptor (CXR) mRNAs were quantified using SYBR Green Realtime PCR Master Mix (Toyobo, Japan) on a real-time PCR detection system (Bio-Rad Laboratories, Hercules, CA, USA). GAPDH was used as an internal control. Relative levels of *Abcb1* and CXR were analyzed using the 2^−ΔΔCt^ method.

### 4.4. In Situ Single-Pass Intestinal Perfusion in Berberine-Treated Broilers

The broilers were divided into three experimental groups and drugs were administered as described above. Broilers were fasted for 12 h with free access to water before the perfusion experiment. Then, the animals were anesthetized with urethane intraperitoneally (0.5 mL/kg b.w.) and an incision was made through the abdominal cavity. A 10-cm intestinal segment was carefully exposed to air for intestinal perfusion. Two polyethylene cannulas were inserted through small slits at the proximal and distal ends (inlet and outlet). Initially, the intestinal segment was gently flushed with Krebs-Ringer buffer for about 30 min until a clear effluent flowed out. Then, Rho123 (1.5 ng/mL) solution was introduced into the loop at a constant flow rate of 0.2 mL/min. The intestinal fluid was collected in bottles at the outlet of the intestine at intervals of 10 min for up to 100 min. The lengths and internal radii of the intestinal segments were measured after all of the chickens were killed by cervical dislocation.

All samples were weighed and centrifuged for 10 min at 4500 rpm. The supernatants were filtered using a 0.22-μm filter (Millipore, Burlington, MA, USA) and samples were collected for analysis by high-performance liquid chromatography (HPLC). The apparent permeability coefficient (*P*_app_) and absorption rate constant (*Ka*) of Rho123 were calculated as follows:(1)$$P_{\mathrm{app}}=\frac{-Q_{\ln}\left(\frac{C_{\mathrm{out}}\cdot Q\mathrm{out}}{C_{\mathrm{in}}\cdot Q_{\mathrm{in}}}\right)}{2\pi rl}$$
(2)$$K_{a}=\left(1-\frac{C_{\mathrm{out}}\cdot Q_{\mathrm{out}}}{C_{\mathrm{in}}\cdot Q_{\mathrm{in}}}\right)\cdot \frac{Q}{V}$$
where *Q*_in_/*Q*_out_ are the intestinal perfusate input and output volumes (mL), *C*_in_ and *C*_out_ represent the mass concentrations of the enteric importer and exporter perfusate (μg/mL), *Q* represents the perfusion rate (0.2 mL/min), *V* is the volume of bowel perfusion, and 2πrl is the area of the mass transfer surface (cm^2^).

### 4.5. Bidirectional Transport Experiments Using MDCK-Abcb1 Cells

To confirm whether berberine affects the function of P-gp and the efficiency of inhibition, the bidirectional transport of Rho123 (a classical P-gp substrate) was evaluated using MDCK-chAbcb1 cells. Verapamil (a potent inhibitor of P-gp) served as a positive control. Cells were seeded in a 12-well Transwell plate (Corning, Acton, MA, USA) at a density of 1.0 × 10^5^ cells/well and cultured for 5 days until the transepithelial electrical resistance values were greater than 300 Ω cm^2^. First, the monolayers were preincubated with Hank’s balanced salt solution either alone or with berberine (5, 20, and 40 μM) or verapamil (10, 50, 100, 1000, and 5000 μM) for 2 h or with berberine (5, 20, and 40 μM) for 8 h. Then, the bidirectional transport of Rho123 was carried out by replacing the donor buffer with transport buffer containing the mixture of Rho123 (4 mM) and the indicated concentrations of berberine or verapamil. The samples were collected at 20, 40, 60, and 80 min from the receiving sides and analyzed by HPLC.

To prove that berberine could inhibit the efflux function of P-gp, bidirectional transport experiments were further conducted following the above procedure using commonly used veterinary drugs, including tilmicosin, ciprofloxacin, clindamycin, sulfadiazine, ampicillin, and enrofloxacin, which were identified as potential substrates. The berberine preincubation time (5, 20, and 40 μM) was 2 h and the transport concentrations of these antimicrobial drugs were all 20 μM. Enrofloxacin and ampicillin were transported for 1 h and the others for 4 h. Then, the samples collected from the receiving sides were subjected to an HPLC analysis. Of note, the transport periods for all of the drugs were determined by a preliminary experiment to avoid the saturation of P-gp.

To evaluate whether berberine is a substrate for P-gp, bidirectional transport experiments using berberine were performed following the above procedure for Rho123 transport. The preincubation time of verapamil (100 μM) was 2 h and the transport concentration of berberine was 100 μM. After transport for 4 h, the samples collected from the receiving sides were used for HPLC.

The apparent permeability coefficients (*P*_app_) and the efflux ratio (ER) were calculated as follows:(3)$$P_{\mathrm{app}}=\frac{dQ}{dt}\times \frac{1}{\left(A\times C_{o}\right)}$$
(4)$$\mathrm{ER}=\frac{P_{\mathrm{app}}\left(B\to A\right)}{P_{\mathrm{app}}\left(A\to B\right)}$$

In the third equation, A is the membrane surface area of the filter, C_0_ is the initial concentration of the test drug, *dQ* is the amount of transported drug, and *dt* is the time elapsed. In the fourth equation, *P*_app_ B→A and *P*_app_ A→B are the apparent permeability coefficients for the basolateral-to-apical and apical-to-basolateral directions, respectively.

The inhibition rate was calculated using the following equation:(5)$$\mathrm{Inhibition}\;\mathrm{rate}=\frac{{\mathrm{ER}}_{\mathrm{control}}-{\mathrm{ER}}_{\mathrm{treatment}}}{{\mathrm{ER}}_{\mathrm{control}}}$$

The transport ratio (r) was calculated by dividing the apically directed translocation (B→A) by the basolaterally directed translocation (A→B) of berberine.

### 4.6. Molecular Modelling

Structures of berberine and several known P-gp substrates or inhibitors (such as morphine, Rho123, indinavir, ritonavir, and saquinavir, selected as controls in molecular docking experiments), were generated from canonical SMILES strings that were obtained from the PubChem database [36]. Avogadro version 1.2.0 [37] was used to set protonation states of ionizable groups to reflect dominant charge states at pH 7.4. A conformation search utilizing MMFF94 implemented in Avogadro was used to generate the lowest energy conformations for all molecules, which were stored in the mol2 file format.

Homology modeling of CXR was performed following a previously described procedure [15]. The homology model for chicken P-glycoprotein (CP-gp) was produced as described previously [15], with minor modifications. Briefly, the GenPept database was used to retrieve the sequence of CP-gp (accession number NP_990225), which was submitted to the I-TASSER homology modeling server [38] to obtain a three-dimensional structure. Careful examination of the results indicated that the top two solution models may not be adequate for docking studies as the N-terminal region was predicted to protrude near the putative binding site. Therefore, a third model was selected for molecular docking. Initially, Maestro Graphic User Interface v2018-1 was used to overlay the third model onto mouse P-gp (PDB ID: 3G60), the top structural analogue. The ligand (PDB ID: 0JZ) was copied into the CP-gp binding site followed by the manual removal of steric clashes. This complex was processed using Protein Preparation Wizard prior to molecular dynamics simulations [39]. The system with all hydrogen atoms and the protonation states of ionizable groups in the protein set corresponding to pH 7 were prepared for further modeling studies using System Builder. The protein and ligand complex were embedded into the 1-palmitoyl-2-oleoyl-glycero-3-phosphocholine (POPC) membrane model and solvated with a box of explicit water molecules represented as single point charges, and ions were added to neutralize the whole system. The size of the box was 10 Å larger than the size of a protein in all directions. Minimization and molecular dynamic simulations were performed using Desmond and OPLS2005 force fields. Each system was initially optimized and then simulated for 1 ns at 300 K and 1 atm, and the docking structures were extracted from the final frame of the molecular dynamics trajectories. The three-dimensional structures of the chicken ATP-binding cassette sub-family C member 7 (CABCC7; accession number: ABK34432) and chicken multi-antimicrobial extrusion protein (CMATE: accession number: NP_001025891) transporters were obtained using a similar procedure but were not embedded into the membrane and were not subjected to molecular dynamic simulations.

The docking of all ligands into the active sites of CXR, CP-gp, CABCC7, and CMATE was evaluated using AutoDock Vina [40] and VegaZZ as a graphical user interface [41]. The geometrical centers of the bound ligands, removed from the structure prior to docking, were used to position the binding sites, with sizes of 24 Å × 24 Å × 24 Å. The top nine binding poses were calculated for all ligands with exhaustiveness set to 20.

The top scoring poses of berberine in the binding pockets of both proteins were selected for molecular dynamics simulations to explore the dynamic nature of protein–berberine interactions. The systems were prepared and simulations were conducted following the methods described above. Both trajectories were analyzed using the “Simulations Interactions Diagram” and “Trajectory Clustering” functions built in Maestro. Cluster representatives from trajectories were re-scored using Adaptive Poisson-Boltzmann Solver (APBS) [42] built in VegaZZ. All images were generated using Biovia Discovery Studio 2016.

### 4.7. Statistical Analysis

Results are presented as the mean ± standard error of the mean (SEM). SPSS (version 20.0, SPSS Inc., Chicago, IL, USA) was used for statistical analyses. One-way analysis of variance and post hoc Tukey’s tests were used to assess the statistical significance of differences. *p* < 0.05 was considered significant and *p* < 0.01 was considered extremely significant.

## 5. Conclusions

In summary, our work demonstrates that berberine could down-regulate the expression and activity of P-gp in broilers, thus increasing the absorption of P-gp substrates both in situ and in vitro. The current results should be helpful in guiding the rational use of drugs in the poultry industry to increase drug bioavailability or avoid possible adverse effects. Further studies are necessary to assess the impact of berberine on P-gp substrate drugs in broilers to pave the way towards clinical applications.

## Figures and Tables

**Figure 1 ijms-20-01966-f001:**
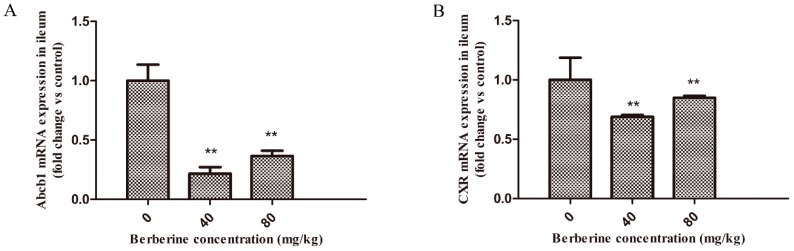
The effect of berberine on *Abcb1* and chicken xenobiotic receptor (*CXR*) mRNA levels in broilers. (**A**) *Abcb1*. (**B**) *CXR*. Data are represented as mean ± SEM of three independent experiments. ** *p* < 0.01.

**Figure 2 ijms-20-01966-f002:**
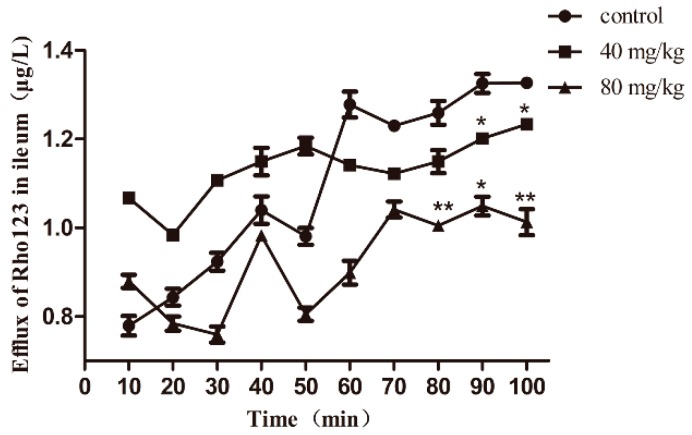
Mean Rho123 concentrations vs. time curves in the jejunum of broilers after oral administration of berberine for 24 h. Each point represents the mean ± SEM of six broilers. * *p* < 0.05; ** *p* < 0.01.

**Figure 3 ijms-20-01966-f003:**
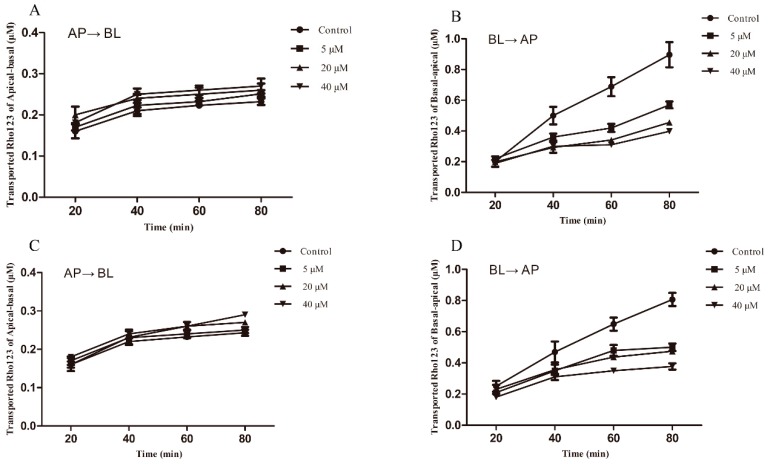
Bi-directional transport of Rho123 across MDCK-chAbcb1 cell monolayer after pre-treatment with berberine for the indicated time. (**A**) 2 h, Apical to Basolateral (AP→BL) (**B**) 2 h, Basolateral to Apical (BL→AP) (**C**) 8 h, Apical to Basolateral (AP→BL). (**D**) 8 h, Basolateral to Apical (BL→AP). Each point represents as mean ± SEM of three independent experiments.

**Figure 4 ijms-20-01966-f004:**
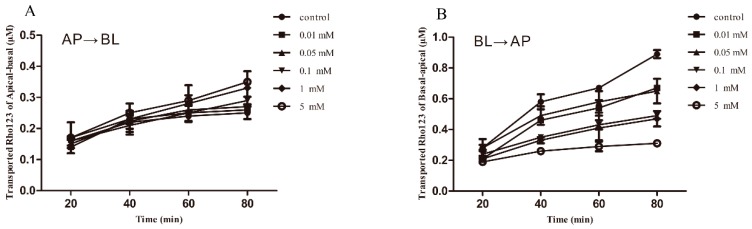
Bi-directional transport of Rho123 across MDCK-chAbcb1 cell monolayer after pre-treatment with verapamil for 2 h. (**A**) Apical to Basolateral (AP→BL). (**B**) Basolateral to Apical (BL→AP). Each point represents as mean ± SEM of three independent experiments.

**Figure 5 ijms-20-01966-f005:**
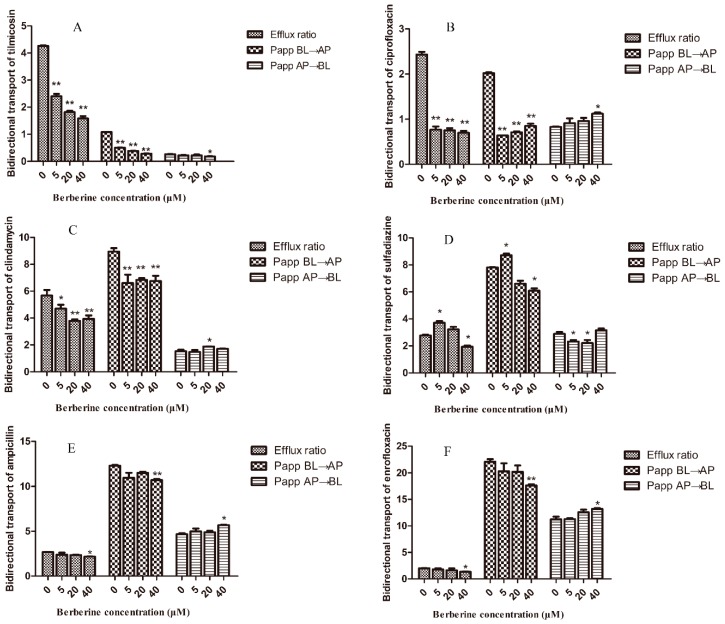
Bi-directional transport of difference chicken P-gp substrates across MDCK-chAbcb1 cell monolayer after pre-treatment with berberine for 2 h. (**A**) Tilmicosin. (**B**) Ciprofloxacin. (**C**) Clindamycin. (**D**) Sulfadiazine. (**E**) Ampicillin. (**F**) Enrofloxacin. Data are shown as mean ± SEM of three independent experiments. * *p* < 0.05; ** *p* < 0.01.

**Figure 6 ijms-20-01966-f006:**
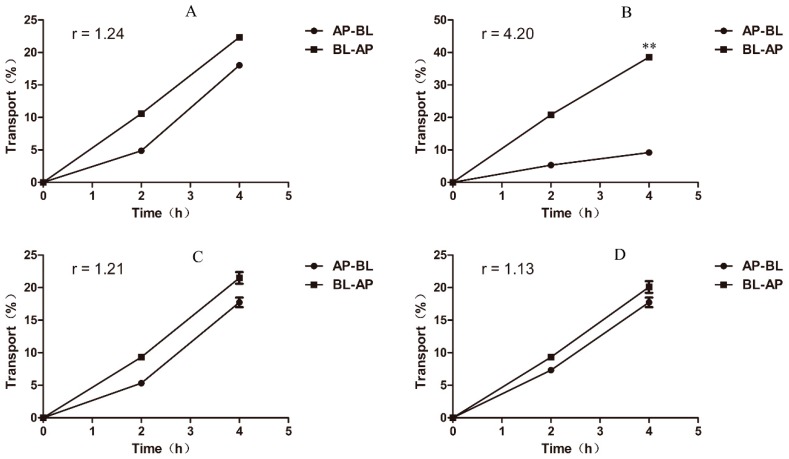
Bi-directional transport of berberine across MDCK and MDCK-chAbcb1 cell monolayers with or without verapamil (**A**) MDCK cell monolayer without verapamil. (**B**) MDCK-chAbcb1 cell monolayer without verapamil. (**C**) MDCK cell monolayer with verapamil. (**D**) MDCK-chAbcb1 cell monolayer with verapamil. Data are shown as mean ± SEM of three independent experiments. ** *p* < 0.01.

**Figure 7 ijms-20-01966-f007:**
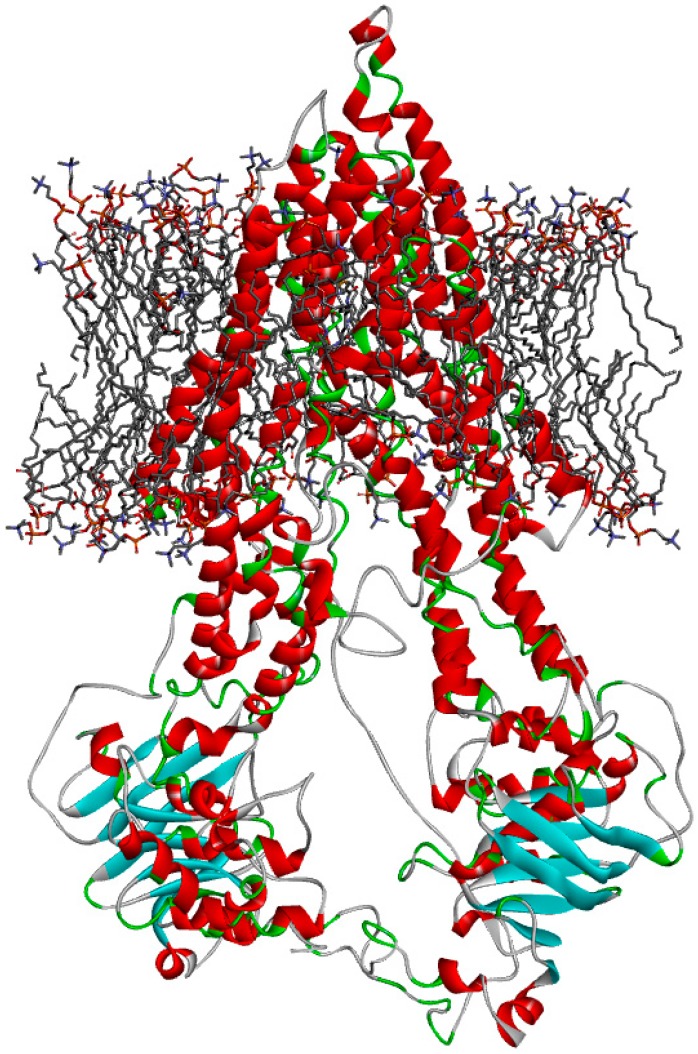
Final snapshot of the 1 ns molecular dynamics simulation of the CP-gp model build by homology modelling using I-TASSER homology modelling server and embedded in the POPC lipid bilayer. CP-gp is shown in the ribbon representation colored according to secondary structure, while the POPC molecules are shown in the stick representation (hydrogen atoms not shown for clarity).

**Figure 8 ijms-20-01966-f008:**
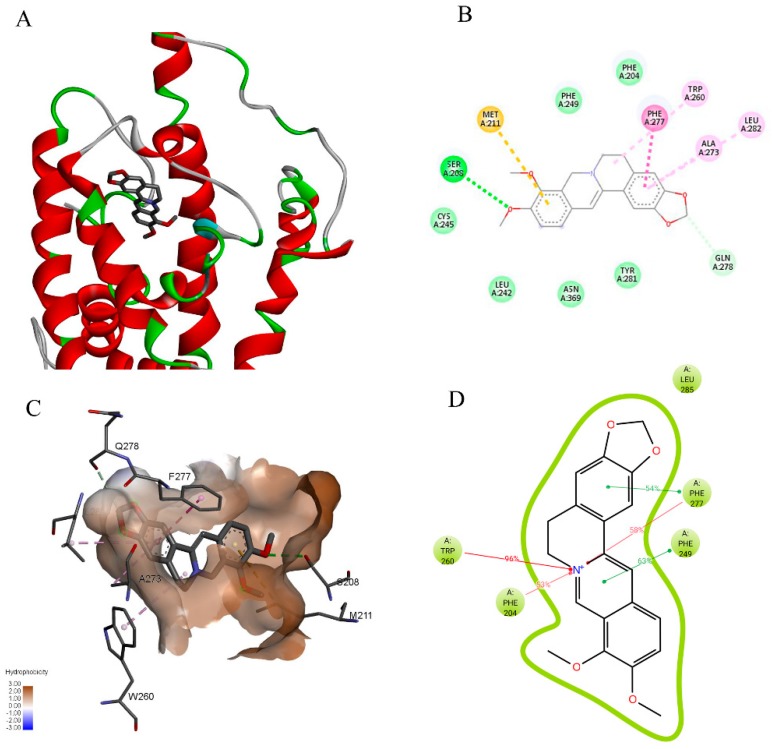
Molecular docking solution of berberine against CXR target obtained using AutoDock Vina. (**A**) Berberine (thick grey lines) in the ligand binding domain of CXR (ribbon representation), (**B**) Ligand interaction diagram of the berberine docking pose (dark green—conventional hydrogen bond; light green—carbon hydrogen bond; dark pink—pi-pi stacked interaction; light pink—pi-alkyl interaction; yellow—pi-sulfur interaction), (**C**) Berberine in the active sites delineated by the hydrophobic surface and surrounding residues which are labelled and represented as thin grey lines. Hydrogen atoms are not shown for clarity. (**D**) Dynamic nature of interactions represented by percentage of frames for which interactions was observed during 1 ns of molecular dynamics simulations of the fully solvated system.

**Figure 9 ijms-20-01966-f009:**
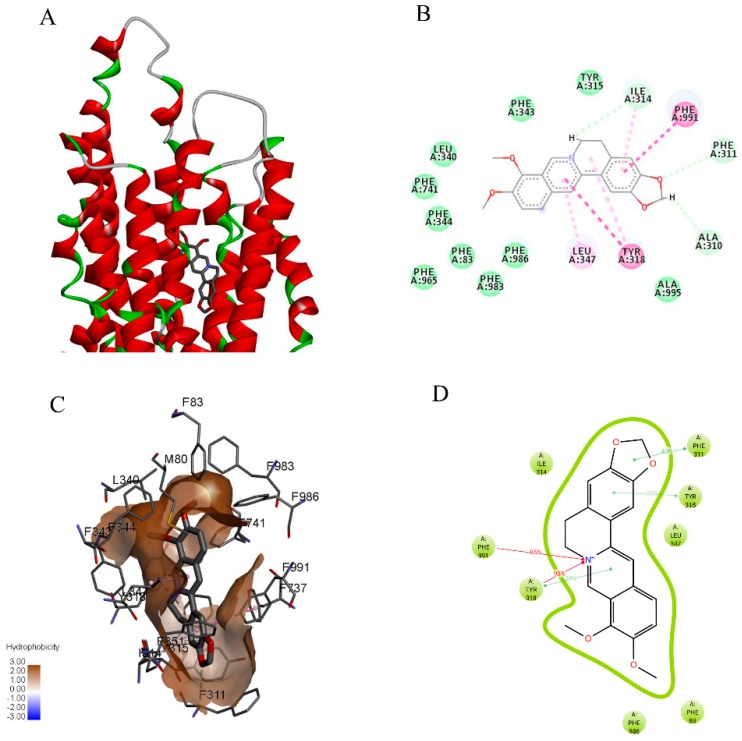
Molecular docking solution of berberine against CP-gp target obtained using AutoDock Vina. (**A**) Berberine (thick grey lines) in the ligand binding domain of CP-gp (ribbon representation), (**B**) Ligand interaction diagram of the berberine docking pose (dark green—conventional hydrogen bond; light green—carbon hydrogen bond; dark pink—pi-pi stacked interaction; light pink—pi-alkyl interaction; yellow—pi-sulfur interaction), (**C**) Berberine in the active sites delineated by the hydrophobic surface and surrounding residues which are labelled and represented as thin grey lines. Hydrogen atoms are not shown for clarity. (**D**) Dynamic nature of interactions represented by percentage of frames for which interactions was observed during 1 ns of molecular dynamics simulations of the fully solvated system.

**Table 1 ijms-20-01966-t001:** The effect of berberine on *K*_a_ and *P*_app_ of Rho123 in chicken ileum.

Parameter	Control	Berberine (24 h)	
		40 mg/kg	80 mg/kg
*K*_a_ (1/min)	1.12 ± 0.28	1.87 ± 0.32 *	2.25 ± 0.29 *
*P*_app_ (×10^−4^ cm/s)	1.23 ± 0.42	2.2 ± 1.41	1.26 ± 0.42

Mean ± SEM (*n* = 6), * *p* < 0.05, compared with control.

**Table 2 ijms-20-01966-t002:** Effect of berberine on the efflux ratio and inhibition rate of Rho123 in MDCK-chAbcb1 cells.

Parameter	Control	Berberine (2 h)			Control	Berberine (8 h)		
		5 μM	20 μM	40 μM		5 μM	20 μM	40 μM
*P*_app_AP→BL(×10^−6^ cm/s)	8.86 ± 1.38	5.43 ± 0.41	6.90 ± 0.76	9.94 ± 1.47	8.62 ± 1.24	6.43 ± 1.01	8.31 ± 0.8	7.35 ± 0.36
*P*_app_BL→AP(×10^−6^ cm/s)	62.9 ± 7.69	23.26 ± 1.77	23.26 ± 1.77 *	18.49 ± 5.88 **	60.1 ± 5.13	27.11 ± 2.27	22.18 ± 2.43 *	17.35 ± 0.82 **
Efflux ratio	7.28 ± 0.98	4.26 ± 0.85	3.4 ± 0.66 *	1.92 ± 0.79 **	7.24 ± 0.49	4.37 ± 0.93	2.71 ± 0.51 *	2.3 ± 0.16 **
Inhibition rate	0	41.48	53.30	73.63	0	39.64	62.57	68.23

Mean ± SEM (*n* = 6), * *p* < 0.05, ** *p* < 0.01, compared with control.

**Table 3 ijms-20-01966-t003:** Effect of verapamil on the efflux ratio and inhibition rate of rhodamine 123 in MDCK-chAbcb1 cells.

Verapamil (µM)	*P*_app_ AP→BL (×10^−6^ cm/s)	*P*_app_ BL→AP (×10^−6^ cm/s)	Efflux Ratio	Inhibition Rate
0	8.04 ± 0.76	54.4 ± 4.34	6.24 ± 0.7	0
10	11.7 ± 0.53	41.1 ± 0.159	4.52 ± 0.11	27.56
50	7.4 ± 0.25	33.4 ± 4.02	4.16 ± 0.866	33.33
100	8.6 ± 1.5	27.7 ± 0.68 *	3.51 ± 0.23 *	43.75
1 000	9.1 ± 0.25	22.8 ± 1.56 *	1.94 ± 0.098 *	68.91
5 000	8.04 ± 0.76	10.7 ± 0.65 **	1.37 ± 0.21 **	78.04

Mean ± SEM (*n* = 6), * *p* < 0.05, ** *p* < 0.01, compared with control.

**Table 4 ijms-20-01966-t004:** Docking scores of berberine compared to selected P-gp substrates and inhibitors obtained using the predicted binding affinities to chicken xenophobic receptor (CXR) and chicken P-glycoprotein (CP-gp).

Name	CXR					CP-gp	
	Docking Score (kcal/mol) ^&^	Rescore *		Docking Score (kcal/mol) ^&^		Rescore *	
		Xscore	DSX			Xscore	DSX
berberine	−7.8	−9.2	−160.8	−9.5		−9.41	−196.9
morphine	−8.1			−9.5			
rhodamine123	−9.8			−9.4			
indinavir	−9.1			−10.3			
ritonavir	−8.8			−10.1			
saquinavir	−10.8			−10.8			

&—docking score obtained using Autodock Vina for all small molecules. *—rescoring using the representative of the most populated cluster from the trajectory.

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
