# Peer review of "Inhibitory Effect of Berberine on Broiler P-glycoprotein Expression and Function: In Situ and In Vitro Studies"

_ijms, 2019, doi:10.3390/ijms20081966_

Round 1
Reviewer 1 Report
The paper deals with interactions of berberine with chicken broiler P-gp (ABCB1) system and chicken xenobiotic receptor describing inhibition of P-gp by berberine. Interestingly, recently, an upregulation of P-gp by berberine was described in human Caco-2 cell model (Jing et al. 2018).Please discuss this discrepancy in detail in the Discussion.
Please add the word "broiler" into the title - maybe ...chicken broiler P-glycoprotein....as the broilers are specific hybrids.
Author Response
Point 1: The paper deals with interactions of berberine with chicken broiler P-gp (ABCB1) system and chicken xenobiotic receptor describing inhibition of P-gp by berberine. Interestingly, recently, an upregulation of P-gp by berberine was described in human Caco-2 cell model (Jing et al. 2018). Please discuss this discrepancy in detail in the Discussion.
Response 1: Thank you for your helpful suggestions. We have discussed the regulatory discrepancy between chicken broiler P-gp and human P-gp in the discussion (line262-270).
Point 2: Please add the word "broiler" into the title - maybe ...chicken broiler P-glycoprotein....as the broilers are specific hybrids.
Response 2: Thank you for your helpful suggestions. We have added the word "broiler" into the title. The revised title is “Inhibitory Effect of Berberine on broiler P-glycoprotein Expression and Function: In Situ and In Vitro Studies”.
Reviewer 2 Report
Paper brings novel data that could be published. I do not comment to the data obtained and their interpretation. I have only minor recommendation. Correct description of Ka and Papp definition given on page twelve as: Ka = (1 – Cout Qout/Cin Qin)Q/V,
Papp = −Qin(Cout Qout/Cin Qin)/2πrl,
by using of equation utility of word processor (MS WORD or other). Moreover if definition of Papp is in your paper correct it could by wrote in simpler expression: Papp=-(Cout Qout/Cin)/2πrl. Is the equation correct?
Author Response
Point 1: Paper brings novel data that could be published. I do not comment to the data obtained and their interpretation. I have only minor recommendation. Correct description of Ka and Papp definition given on page twelve as:
Ka = (1 – Cout Qout/Cin Qin)Q/V,
Papp = −Qin(Cout Qout/Cin Qin)/2πrl,
by using of equation utility of word processor (MS WORD or other). Moreover if definition of Papp is in your paper correct it could by wrote in simpler expression: Papp=-(Cout Qout/Cin)/2πrl. Is the equation correct?
Response 1: Thank you for your appreciation and helpful comments. First of all, I'm very sorry to make such a mistake due to my careless. The correct equation is as follows:
where the “ln” represents the logarithmic function. We have corrected all the equations in our paper (line 363、393 and 399).